# Variational Iteration Method for Solving Fractional Integro-Differential Equations with Conformable Differointegration

**Mondher Damak [1] and Zaid Amer Mohammed [2,\*]**

[1] Faculty of Sciences of Sfax, University of Sfax, P.O. Box 1171, Sfax 3000, Tunisia
[2] Economic and Administration College, Al-Iraqia University, Baghdad 10071, Iraq
\* Correspondence: zaid.amer84@yahoo.com

**Abstract:** Multidimensional integro-differential equations are obtained when the unknown function of several independent variable and/or its derivatives appear under an integral sign. When the differentiation or integration operators or both are of fractional order, the integral equation in this case is called a multidimensional fractional integro-differential equation. Such equations are difficult to solve analytically; therefore, as the main objective of this paper, an approximate method—which is the variational iteration method—will be used to solve this type of equation with conformable fractional-order derivatives and integrals. First, we drive the iterative sequence of approximate solutions using the proposed method, and then, under certain conditions over the kernel of the integro-differential equation, prove its convergence to the exact solution. Two illustrative examples, linear and nonlinear, are given, and their approximated solutions are simulated using computer programs in order to verify from the reliability and applicability of the proposed method.

**Keywords:** multidimensional integro-differential equations; conformable fractional diffrointegrations; variational iteration method; convergence of the iterative method; general Lagrange multiplier

## 1. Introduction

Recently, fractional differential equations, fractional calculus and fractional differential integral equations became highly important in several branches of science and engineering because many mathematical models are used to formulate different phenomena, such as mechanics, physics, chemical kinetics, astronomy, biology, economics, potential theory and electrostatistics, which are modeled using integro-differential equations [1–6]. In 2016, Shi [7] introduced a formula of mild solutions for impulsive fractional evolution equations.

Many academic researchers continue to be interested in the use of fractional differential equations and/or integral equations, which are based on the development and applications of fractional calculus [8,9]. Nonlinear fractional integral equations and integro-differential equations are notoriously difficult to solve analytically. Moreover, accurate solutions for these equations are extremely rare. As a result, various authors have taken an interest in numerically solving these problems, particularly after the big revolution in computer application. Among the techniques used to solve integro-differential equations are the multi-step methods [10], Adomian decomposition method (ADM) [11], homotopy perturbation method (HPM) [12–14], homotopy analysis method (HAM) [15], variational iteration method (VIM) [16] and so on.

The VIM has been successfully applied to solve many problems in different fields of mathematics and its applications. For example, He was the first researcher to propose the use of VIM to solve linear and nonlinear differential and integral equations [17]. In 1998, He used VIM to solve the classical Blasius equation, ref. [18] and in 1999, he provided the approximate solutions for some well-known nonlinear problems [19]. In 2000, He used the VIM to solve autonomous ordinary differential equations. Moreover, in 2006, Soliman applied the VIM to solve the kdv-Burger's and Lax's seventh-order kdv equations. In the

same year, VIM was applied by Abulwafa and Momani [20] to solve a nonlinear coagulation problem with mass loss and Odibat et al. employed the VIM to solve nonlinear differential equations of fractional order in 2006. In 2006, the VIM was also utilised by Bildiki et al. to solve a variety of problems, including nonlinear partial differential equations, Dehghan and Tateri's Fokker–Plank equation and quadratic Riccati differential equations with constant coefficients. Wang [21] used the VIM to solve integro-differential equations in 2009, while Sweilam used the VIM to solve both linear and nonlinear boundary value problems of the fourth-order integro-differential equations.

In 2009, Wen-Hua Wang used the VIM to solve certain types of fractional integro-differential equations [21]. In 2011, Muhammet and Adin used the VIM to solve the problem of nonlinear fractional integro-differential equations [20].

If an exact solution exists, the VIM provides rapidly convergent consecutive approximations to the precise solution; otherwise, a few approximations might be employed for numerical results.

In this paper, we shall present the VIM used to solve integro–differential equations with conformable fractional order differointegration of the form:

$$T_x^\alpha u(x,y) = g(x,y) + I_x^\beta I_y^\gamma K(x,y,s,t,u(x,y)) \tag{1}$$

where $K$ is given continuous function, $0 < \alpha \le 1$, $\beta$, $\gamma > 0$, $x,y \in [a,b] \times [c,d]$ here, $T^\alpha$ is understood as a conformable fractional derivative of order $\alpha$, while $I^\beta$ and $I^\gamma$ stands for conformable fractional-order integrals of order $\beta$ and $\gamma$, respectively.

## 2. Main Concepts of Factional Calculus

Among the most important definitions of fractional-order derivatives or integrals which will be used next in this paper is the conformable type, which is more simple than other definitions and more stable in comparison with the nonfractional (or integer order) derivatives and integrals.

**Definition 1** (**Conformable Fractional-Order Derivative** [22])**.** *Given a function $f : [a, \infty) \to \mathbb{R}$, then the left conformable fractional derivative of order $\alpha$ can be defined as:*

$$(T_\alpha^a(f))(x) = \lim_{\varepsilon \to 0} \frac{f\left(x + \varepsilon(x-a)^{1-\alpha}\right) - f(x)}{\varepsilon}$$

*for all $x > 0, \alpha \in (0,1]$. When $a = 0$, we write $T_\alpha$. If $(T_\alpha^a(f))(x)$ exist on the interval $(a,b)$, then define:*

$$(T_\alpha^a(f))(a) = \lim_{x \to a^+} f^{(\alpha)}(x)$$

*The right conformable fractional derivative of order $\alpha \in (0,1]$ terminating at $b$ of $f$, is defined by:*

$$_\alpha^b T(f)(x) = \lim \frac{f\left(x + \varepsilon(b-x)^{1-\alpha}\right) - f(x)}{\varepsilon}$$

*If $_\alpha^b(T(f)(x))$ exist on the interval $(a,b)$, then define:*

$$_\alpha^b(T(f))(b) = \lim_{x \to b^-} {}^b T_\alpha(f)(x)$$

**Definition 2** (**Conformable Fractional-Order Integral** [23])**.** *Given a continuous function $f : [a, \infty) \to R$, then the left conformable fractional integral of order $\alpha$ of $f$ is:*

$$(I_\alpha^a(f))(x) = \int_a^x f(s)d\alpha(s,a) = \int_a^x \frac{f(s)}{(s-a)^{1-\alpha}} ds \tag{2}$$

*where the integral is considered as the usual Riemann improper integral and $a \geq 0$. On the other hand, in the right case, we have:*

$$\,^b_\alpha I_x(f)(x) = \int_x^b f(s) d\alpha(b,s) = \int_x^b \frac{f(s)}{(b-s)^{1-\alpha}} ds \tag{3}$$

Among some different properties concerning fractional integrals and derivative which are very useful in applications are the following, where $T^\alpha$ and $I^\alpha$ refer to fractional-order conformable and integral, respectively:

1. $T^\alpha(c_1 f + bc_2) = c_1 T^\alpha(f) + c_2 T^\alpha(g)$, for all $c_1, c_2 \in \mathbb{R}$.
2. $T^\alpha(c_1 f + c_2 g) = c_1 T^\alpha f + c_2 T^\alpha g$.
3. $T^\alpha(c_1 f + bc_2) = c_1 T^\alpha(f) + c_2 T^\alpha(g)$, for all $c_1, c_2 \in \mathbb{R}$.
4. $T^\alpha(x^p) = px^{p-\alpha}$, for all $p \in \mathbb{R}$.
5. $T^\alpha(\lambda) = 0$, for all constant functions $f(x) = \lambda$.
6. $T^\alpha(fg) = fT^\alpha(g) + gT^\alpha(f)$
7. $T^\alpha\left(\frac{f}{g}\right) = \frac{gT^\alpha(f) - fT^\alpha(g)}{g^2}, g \neq 0$.
8. If, in addition, $f$ is differentiable, then $T^\alpha(f)(x) = x^{1-\alpha}\frac{df}{dt}(x)$.
9. $T^a_{\alpha x} I^a_\alpha(f)(x) = f(x)$, for $x > a$, where $f$ is any continuous function in the domain of $I^a_\alpha$ and $\,^b_\alpha T^b_\alpha I_x(f)(x) = f(x)$, for $x < b$, where $f$ is any continuous function in the domain of $\,^b_\alpha I$.

## 3. Variational Iteration Method

The essential aspect of the VIM, as previously stated in the literature, is that the solution of a mathematical problem under the linearization assumption is utilized as a starting approximation or trial function for the next successive approximate solution to the problem under certain conditions [2].

Consider the following general nonlinear equation in operator form to demonstrate the VIM's essential concept [3]:

$$Au(x) = g(x) \tag{4}$$

and suppose that Equation (1) may be decomposed as:

$$L(u(x)) + N(u(x)) = g(x), x \in [a, b] \tag{5}$$

where A is any operator that may be decomposed into linear and nonlinear operators $L$ and $N$, respectively, and $g(x)$ is any function that is referred to as the nonhomogeneous term. Equation (5) may be solved iteratively using the VIM by using the correction functional defined by:

$$u_{n+1}(x) = u_n(x) + \int_{x_a}^a \lambda(x,s)\{L(u_n(s)) + N(\tilde{u}_n(s)) - g(s)\}ds, n = 0, 1, \ldots \tag{6}$$

where $\lambda$ is the general Lagrange multiplier that may ideally be discovered using variational theory the *nth* approximation of the subscript $n$ denotes the solution $u$, and $\tilde{u}_n$ is considered a restricted variation, i.e., $\delta\tilde{u}_n = 0$ where the $\delta$ is the first variation [3].

## 4. Applications of the Vim for Multidimensional Integro-Differential Equations of Fractional Order

Consider the fractional integro-differential Equation (1), which may be written as:

$$T^\alpha_x u(x,y) - I^\beta_x I^\gamma_y K(x,y,s,t,u(s,t)) - g(x,y) = 0 \tag{7}$$

Multiplying Equation (7) by a general Lagrange multiplier $\lambda$, yields to:

$$\lambda(s,t)\{T^\alpha_x u(x,y) - I^\beta_x I^\gamma_y K(x,y,s,t,u(s,t)) - g(x,y)\} = 0 \tag{8}$$

Now, take $I_x^\alpha$ to both sides of Equation (8), which give:

$$I_x^\alpha[\lambda(s,t)\{T_x^\alpha u(x,y) + I_x^\beta I_y^\alpha K(x,y,s,t,u(s,t))\} - g(x,y)] = 0 \tag{9}$$

Then, the correction functional with respect to $x$ will be read as follows:

$$u_{n+1}(x,y) = u_n(x,y) + I_x^\alpha[\lambda(w,y)\{T_w^\alpha u(w,y) - I_w^\beta I_y^\gamma K(w,y,s,t,u(s,t)) - g(w,y)\}] = 0 \tag{10}$$

and the problem now is to evaluate $\lambda$. The problem of evaluating $\lambda$ is difficult, since Equation (10) consists of functional derivatives and integrals, so to avoid this difficulty, approximate $I_x^\alpha$ and $T_w^a$ for $0 < \alpha \leqslant 1$ by the first integral and derivative. Hence:

$$u_{n+1}(x,y) = u_n(x,y) + \int_a^x \left[\lambda\left(x,y\right)\left(\frac{\partial u_n}{\partial w}(w,y) - g(w,y) - I_w^\beta I_y^\gamma k(w,y,s,t,\widetilde{u_n}(s,t))\right)\right] dw \tag{11}$$

Now, taking the first variation of Equation (11) with respect to $u_n$, give:

$$\delta u_{n+1}(x,y) = \delta u_n(x,y) + \delta \int_{x_0}^x \left[\lambda\left(\frac{\partial u_n}{\partial s} - g - I_x^\beta I_y^y k(s,t,u_n(s,t))\right)\right] dw \tag{12}$$

$$= \delta u_n(x,y) + \delta \int_{x_a}^x \lambda(w,y)\left(\frac{\partial u_n(w,y)}{\partial w}\right) dw \tag{13}$$

$$= \delta u_n(x,y) + \int_a^x \lambda(w,y)\delta\left(\frac{\partial u_n(w,y)}{\partial w}\right) dw \tag{14}$$

Using integration by parts,

$$\delta u_{n+1}(x,y) = \delta u_n(x,y) + \delta u_n(w,y)|_{w=x} - \int \frac{\partial \lambda(w,y)}{\partial w}\delta u_n dw \tag{15}$$

$$= (1+\lambda)\delta u_n(w,y)|_{w=x} - \int \frac{\partial \lambda}{\partial w}\delta u_n(w,y) ds \tag{16}$$

$$\frac{\partial \lambda}{\partial w}(w,y)|_{w=x} = 0 \tag{17}$$

$$1 + \lambda(w,y)|_{w=x} = 0 \tag{18}$$

Solving this equation will give $\lambda(w,y) = -1$, and substituting in Equation (10), we get:

$$u_{n+1}(x,y) = u_n(x,y) - I_x^\alpha\left[T_w^\alpha u_n(w,y) - g(w,y) - I_w^\beta I_y^\gamma K\left(w,y,s,t,u_n(s,t)\right)\right] \tag{19}$$

**Theorem 1.** *Let $u, u_n \in C_x^m([a,b] \times [c,d])$, which is a Banach space with a m-th-order continuous partial derivative with respect to $x$, be the exact and approximate solutions of the integro-differential equation of fractional order (1). If $E_n(x,y) = u_n(x,y) - u(x,y)$ and the kernel K satisfy Lipschitz with respect to u, with constant L satisfying*

$$L < \left[\frac{\theta(2\theta+1)(3\theta+1)\ldots(n\theta+1)\gamma(2\gamma+1)(3\gamma+1)\ldots(n\gamma+1)}{(b-a)^{n\theta+1}(d-c)^{n\gamma+1}}\right]^{1/n}$$

*then, the sequences solutions of approximation $\{u_n\}$ converge to the exact solution u.*

**Proof.** Consider the integro-differential equation of fractional order:

$$T_x^\alpha u(x,y) = g(x,y) + I_x^\beta I_y^\gamma K(x,y,s,t,u(s,t)), \text{ where } u(0,y) = u_0$$

The approximate solution using the VIM is given by:

$$u_{n+1}(x,y) = u_n(x,y) - I_x^\alpha \{T_w^\alpha u_n(w,y) - g(w,y) - I_w^\beta I_y^\gamma k(x,y,s,t,u(s,t))\} \tag{20}$$

Since $u$ is the exact solution of the integro-differential of the fractional order,

$$u(x,y) = u(x,y) - I_x^\alpha \{T_w^\alpha u(w,y) - g(x,y) - I_w^\beta I_y^\gamma k(x,y,s,t,u(s,t))\} \tag{21}$$

Then, subtracting Equation (21) from Equation (20), we get:

$$E_{n+1}(x,y) = E_n(x,y) - I_x^\alpha T_w^\alpha u_n(w,y) - g(x,y) + g(x,y) \tag{22}$$
$$-I_w^\beta I_y^\gamma k u_n(x,y,s,t,u_n(s,t)) - T_w^\beta u(w,y) + I_w^\beta I_y^\gamma K(x,y,s,t,u(s,t) + g(x,y))$$

$$= E_n(x,y) - I_x^\alpha \{T_w^\beta E_n(w,y) - I_w^\beta I_y^\gamma \{K(w,y,s,t,u_n(s,t)) - K(x,y,s,t,u_n(s,t))\}$$

From property (3), $I_x^\alpha I_w^\alpha E_n(x,y) = E_n(x,y) - E_n(0,y)$ and since $E_n(0,y) = 0$, then

$$E_{n+1}(x,y) = E_n(x,y) - E_n(x,y) + I_x^\alpha I_x^\beta I_y^\gamma [K(x,y,s,t,u_n(s,t)) - K(x,y,s,t,u_n(s,t))]$$

If $\theta = \alpha + \beta$, then:

$$E_{n+1}(x,y) = I_x^\theta I_y^\gamma [K(x,y,s,t,u_n(s,t)) - K(x,y,s,t,u(s,t))] \tag{23}$$

Now, taking the supermum norm for both sides of Equation (23)

$$\left\| E_{n+1}(x,y) \right\| \le I_x^\theta I_y^\gamma \| K(x,y,s,t,u_n(s,t) - K(x,y,s,t,u(s,t) \|$$

where the $K$ (kernel function) satisfies the Lipschitz condition with constant $L$, then:

$$\left\| E_{n+1}(x,y) \right\| \le L I_x^\theta I_y^\gamma \| u_n - u \|$$

$$= L I_x^\theta I_y^\gamma \| E_n(x,y) \| \tag{24}$$

Using the conformable definition of integrals in Equation (24) implies:

$$\left\| E_{n+1}(x,y) \right\| \le L \int_a^x (s-a)^{\theta-1} \int_c^y (t-c)^{\gamma-1} \| E_n(s,t) \| ds dt$$

$$= L \int_c^x \int_a^y (s-a)^{\theta-1} (t-c)^{\gamma-1} \| E_n(x,y) \| ds dt \tag{25}$$

Now, applying mathematical induction over the last inequality:
If $n = 0$

$$\| E_1(x,y) \| \le L \int_a^x \int_c^y (s-a)^{\theta-1} (t-c)^{\gamma-1} \| E_0(x,y) \| ds dt$$

$$= L \left. \frac{(s-a)^\theta}{\theta} \right|_a^x \left. \frac{(t-c)^\gamma}{\gamma} \right|_c^y \| E_0(x,y) \|$$

$$\le L \frac{(x-a)^\theta}{\theta} \frac{(y-c)^\gamma}{\gamma} \| E_0(x,y) \|$$

If $n = 1$, then:

$$\| E_2(x,y) \| \le L \int_a^x \int_c^y (s-a)^{\theta-1} (t-c)^{\gamma-1} \| E_1(x,y) \| ds dt$$

$$\leq L \int_a^x \int_c^y (s-a)^{\theta-1}(t-c)^{\gamma-1} L \frac{(s-a)^\theta}{\theta} \frac{(t-c)^\gamma}{\gamma} \| E_0(x,y) \| \ dsdt$$

$$= \frac{L^2}{\theta\gamma} \int_a^x \int_c^y (s-a)^{2\theta}(t-c)^{2\gamma-1} \| E_0(x,y) \| dsdt$$

$$= \frac{L^2}{\theta\gamma} \frac{(s-a)^{2\theta+1}}{2\theta+1} \Big|_a^x \frac{(t-c)^{2\gamma+1}}{2\gamma+1} \Big|_c^y \| E_0(x,y) \|$$

$$= \frac{L^2}{\theta(2\theta+1)\gamma(2\gamma+1)} (x-a)^{2\theta+1}(y-c)^{2\gamma+1} \| E_0(x,y) \|$$

If $n = 2$

$$\| E_3(x,y) \| \leq L \int_a^x \int_c^y (s-a)^{\theta-1}(t-c)^{\gamma-1} \| E_2(x,y) \| dsdt$$

$$\leq L \int_a^x \int_c^y (s-a)^{\theta-1}(t-c)^{\gamma-1} \frac{L^2}{w(2\theta+1)\gamma(2\gamma+1)} (s-a)^{2\theta+1}(t-c)^{2\gamma+1} \| E_0(x,y) \| dsdt$$

$$= \frac{L^3}{\theta(2\theta+1)\gamma(2\gamma+1)} \int_a^x \int_c^y (s-a)^{3\theta}(t-c)^{3\gamma} \| E_0(x,y) \| dsdt$$

$$= \frac{L^3}{\theta(2\theta+1)\gamma(2\gamma+1)} \frac{(s-a)^{3\theta+1}}{3\theta+1} \Big|_a^x \frac{(t-c)^{3\gamma+1}}{3\gamma+1} \Big|_c^y \| E_0(x,y) \|$$

$$= \frac{L^3}{\theta(2\theta+1)(3\theta+1)\gamma(2\gamma+1)(3\gamma+1)} (x-a)^{3\theta+1}(y-c)^{3\gamma+1} \| E_0(x,y) \|$$

Hence, by induction, we have:

$$\| E_{n+1}(x,y) \| \leq$$

$$\frac{L^n}{\theta(2\theta+1)(3\theta+1)\ldots(n\theta+1)\gamma(2\gamma+1)(3\gamma+1)\ldots(n\gamma+1)}$$
$$(x-a)^{n\theta+1}(y-c)^{n\gamma+1} \| E_0(x,y) \|$$

and upon taking the supremum values of $x$ and $y$, over $[a,b] \times [c.d]$, getting:

$$\| E_{n+1}(x,y) \| \leq$$

$$\frac{L^n}{\theta(2\theta+1)(3\theta+1)\ldots(n\theta+1)\gamma(2\gamma+1)(3\gamma+1)\ldots(n\gamma+1)}$$
$$(b-a)^{n\theta+1}(d-c)^{n\gamma+1} \| E_0(x,y) \|$$

Since

$$\frac{L^n(b-a)^{n\theta+1}(d-c)^{n\gamma+1}}{\theta(2\theta+1)(3\theta+1)\ldots(n\theta+1)\gamma(2\gamma+1)(3\gamma+1)\ldots(n\gamma+1)} < 1$$

Because

$$L < \left[ \frac{\theta(2\theta+1)(3\theta+1)\ldots(n\theta+1)\gamma(2\gamma+1)(3\gamma+1)\ldots(n\gamma+1)}{(b-a)^{n\theta+1}(d-c)^{n\gamma+1}} \right]^{1/n}$$

Hence as $as \ \ n \to \infty$, we have $\| E_n(x,y) \| \to 0$ , i.e., $u_n(x,y) \to u(x,y) \ as \ n \to \infty$. $\square$

## 5. Illustrative Examples

Two examples of using the VIM to solve linear and nonlinear integro-differential equations with conformable fractional order differo-integration are presented in this section.

**Example 1.** *Consider the linear integral equation of fractional order:*

$$T_x^\alpha u(x,y) = g(x,y) + I_x^\beta I_y^\gamma [(x-y)u(x,y)]$$

*where* $g(x,y) = T_x^\alpha u(x,y) - I_x^\beta I_y^\gamma [(x-y)u(x,y)]$.
*The exact solution is given for comparison purpose by* $u_e(x,y) = x^3 y$
*Hence, starting with the initial guess solution:*

$$u_0(x,y) = g(x,y) = 3yx^{3-\alpha} - \frac{x^{\beta+4}}{\beta+4} \cdot \frac{y^{\gamma+1}}{\gamma+1} + \frac{x^{\beta+3}}{\beta+3} \cdot \frac{y^{\gamma+2}}{\gamma+2}$$

*then, to find* $u_1(x,y)$, *if* $n = 1$, *getting:*

$$u_1(x,y) = u_0(x,y) - I_x^\alpha [T_x^\alpha u_0(x,y) - g(x,y) - I_x^\beta I_y^\gamma [(x-y)u(x,y)]$$

$$
\begin{aligned}
&= 3yx^{2-\alpha} - \frac{x^{\beta+4}}{\beta+4} \cdot \frac{y^{\gamma+1}}{\gamma+1} + \frac{x^{\beta+3}}{\beta+3} \\
&\cdot \frac{y^{\gamma+2}}{\gamma+2} - I_x^\alpha [(9-3\alpha)y^{3-2\alpha} - x^{\beta+4-\alpha} \cdot \frac{y^{\gamma+1}}{\gamma+1} + x^{\beta+3-\alpha} \cdot \\
&\frac{y^{\gamma+2}}{\gamma+2} - 3yx^{2-\alpha} - \frac{x^{\beta+4}}{\beta+4} \cdot \frac{y^{\gamma+1}}{\gamma+1} + \frac{x^{\beta+3}}{\beta+3} \cdot \frac{y^{\gamma+2}}{\gamma+2} - I_x^\beta I_y^\gamma \\
&\left[ 3yx^{2-\alpha} + \frac{x^{\beta+5}}{\beta+4} \cdot \frac{y^{\gamma+1}}{\gamma+1} + \frac{x^{\beta+4}}{\beta+3} \cdot \frac{y^{\gamma+2}}{\gamma+2} + 3y^2 x^{3-\alpha} + \frac{x^{\beta+4}}{\beta+4} \cdot \frac{y^{\gamma+2}}{\gamma+1} - \frac{x^{\beta+3}}{\beta+3} \cdot \frac{y^{\gamma+3}}{\gamma+2} \right]
\end{aligned}
\tag{26}
$$

*and carrying out recursively fractional-order integrals of the order* $\gamma$ *represent to* $y$, *of order* $\beta$ *with respect to* $x$ *and of order of* $\alpha$ *represent to* $y$, *we get the final form of* $u_1(x,y)$, *which is as follows: when* $\alpha = 0.8$ , $\beta = 0.5$ *and* $\gamma = 0.75$ *substitution in Equation* (26).
*We get the following result of* $u_1(x,y)$ *approximate up to six decimals*

$$u_1(x,y) \cong 0.024161885 x^{4.3} y^{2.75} +$$

$$1.0x^{3.0}y + 0.1029601 x^{4.5} y^{1.75} - 0.0012449424 x^{6.8} y^{2.5} - 0.023959269 x^{5.3} y^{1.75}$$

$$-0.11544012 x^{3.5} y^{2.75} + 0.0022746821 x^{5.8} y^{3.5} + 2.1684043 e^{-19} x^{2.2} y - 0.0012025012 x^{4.8} y^{4.5}$$

*Similarly to the calculations* $u_1(x,y)$, *we may find new approximation solution up to three iterations, which are found to be*

$$u_2(x,y) = 9.09495e - 10x^{4.3}y^{2.75} - 0.000199751 x^{5.6} y^{4.5} + 1.0x^{3.0}y$$

$$-0.00000507162 x^{9.1} y^{3.25} + 2.96859e - 9x^{4.5} y^{1.75} + 0.00100941 x^{6.8} y^{2.5} +$$

$$5.67525e - 10x^{5.3} y^{1.75} + 0.00000595113 x^{6.1} y^{6.25} - 0.0000148071 x^{7.1} y^{5.25} + 4.5693e -$$

$$9x^{3.5} y^{2.75} - 0.000185443 x^{7.6} y^{2.5} - 0.00215173 x^{5.8} y^{3.5} + 0.000359167 x^{6.6} y^{3.5} +$$

$$0.0000140055 x^{8.1} y^{4.25} + 9.31323e - 10x^{2.2}y +$$

$$0.00133611 x^{4.8} y^{4.5}$$

$$u_3(x,y) = 0.0241619 x^{4.3} y^{2.75} + 1.0x^{3.0}y + 2.00089e - 11x^{6.8} y^{2.5} - 0.0239593$$

$$x^{5.3} y^{1.75} + 9.41681e - 12x^{3.5} y^{2.75} + 6.36646e - 12x^{5.8} y^{3.5} - \left( -0.386473 x^{2.3} y^{0.75} + 0.879121 x^{1.3} y^{1.75} \right).$$

$$(0.00000525264x^{8.1}y^{3.25}+4.0829e-11x^{5.8}y^{2.5}-7.91503e-7x^{8.9}y^{3.25}$$
$$+0.0379687x^{4.3}y^{1.75}+0.00000787187x^{6.1}y^{5.25}-0.0000113188x^{7.1}y^{4.25}+$$
$$5.946e-12x^{6.6}y^{2.5}+6.78554e-11x^{4.8}y^{3.5}-9.03963e-7x^{6.9}y^{5.25}+$$
$$0.00000150668x^{7.9}y^{4.25}-3.86567e-8x^{8.4}y^{6}+9.63085e-$$
$$12x^{5.6}y^{3.5}+3.465e-8x^{9.4}y^{5}-1.26994e-8x^{10.4}y^{4}+$$
$$1.3112e-20x^{3.5}y^{1.75}+1.74071e-8x^{7.4}y^{7})+9.31323e-10x^{2.2}y+4.77485e-$$
$$11x^{4.8}y^{4.5}$$

*Figure 1 shows the comparison between the exact and the approximated solution for different values of y.*

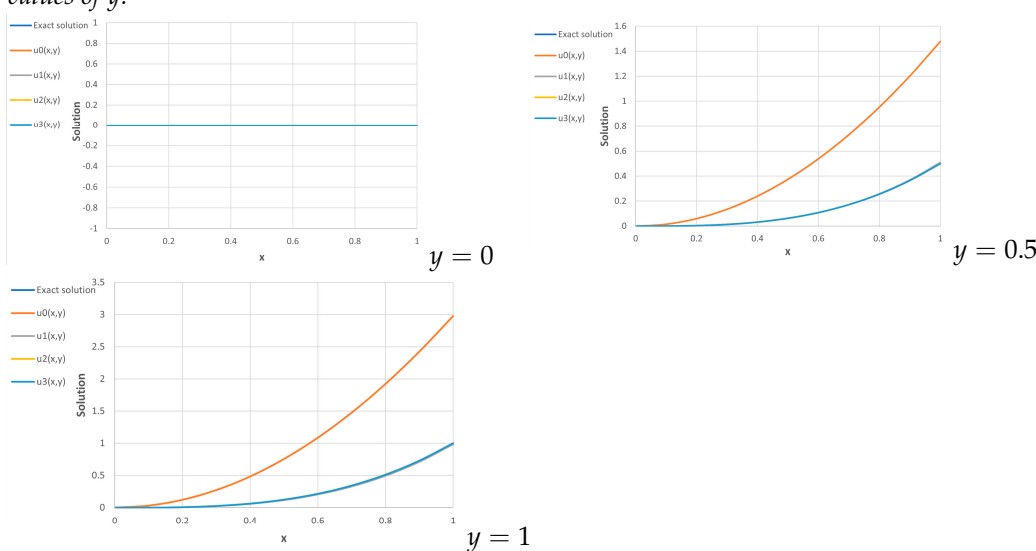

**Figure 1.** Exact and approximate solutions of Example 1.

**Example 2.** *Let the non-linear equation is:*

$$T_x^{\alpha}u(x,y)=g(x,y)+I_x^{\beta}I_y^{\gamma}\left[(xy)e^{u(x,y)}\right]$$

*When the exact solution $u_e(x,y)=x^2y^2$.*

*For the simplicity of calculations and in order to use the properties of conformable and integrations $e^u$ by the Tayler series*

$$e^u=1+u+\frac{u^2}{2!}+\ldots$$

*After tow terms, we have $e^u\cong1+u$ .*

*By calculus of variation, the initial condition to*

$$u_0(x,y)=g(x,y)\ =2y^2x^{2-\alpha}-\frac{x^{\beta+1}}{\beta+1}.\frac{y^{\gamma+1}}{\gamma+1}+\frac{x^{\beta+3}}{\beta+3}.\frac{y^{\gamma+3}}{\gamma+3}$$

*Now, using the variation iteration method to find the next approximation solution as follows:*

$$u_1(x,y)=u_0(x,y)-I_x^{\alpha}[T_x^{\alpha}u_0(x,y)-g(x,y)-I_x^{\beta}I_y^{\gamma}[xy(1+u_0(x,y)]$$

$$= 2y^2 x^{2-\alpha} - \frac{x^{\beta+1}}{\beta+1} \cdot \frac{y^{\gamma+1}}{\gamma+1} + \frac{x^{\beta+3}}{\beta+3}$$

$$\cdot \frac{y^{\gamma+3}}{\gamma+3} - I_y^\alpha [T_x^\alpha 2y^2 x^{2-\alpha} - \frac{x^{\beta+1}}{\beta+1} \cdot \frac{y^{\gamma+1}}{\gamma+1} + \frac{x^{\beta+3}}{\beta+3} \cdot$$

$$\frac{y^{\gamma+3}}{\gamma+3}] - [2y^2 x^{2-\alpha} - \frac{x^{\beta+1}}{\beta+1} \cdot \frac{y^{\gamma+1}}{\gamma+1} + \frac{x^{\beta+3}}{\beta+3} \cdot$$

$$\frac{y^{\gamma+3}}{\gamma+3}] - I_x^\beta I_y^\gamma \left[ xy + 2y^3 x^{1-\alpha} - \frac{x^{\beta+2}}{\beta+1} \cdot \frac{y^{\gamma+2}}{\gamma+1} + \frac{x^{\beta+4}}{\beta+3} \cdot \frac{y^{\gamma+4}}{\gamma+3} \right] \quad (27)$$

*Carrying out recursively order integrals of order $\gamma$ to represent $y$, of order $\beta$ to represent $x$ and of order $\alpha$ to represent $y$, we get the final from of $u_1(x,y)$, which is as follows.*

*When $\alpha = 0.8$ , $\beta = 0.5$ and $\gamma = 0.75$ substitution in Equation (27)*

*We get the following result of $u_1(x,y)$, which is approximate up to six decimals*

$$u_1(x,y) = 1.81826e - 8x^{5.5}y^{5.75} - 0.000477683x^{5.8}y^{5.5}$$

$$-0.0177187x^{4.3}y^{3.75} + 1.0x^{2.0}y^2 - 0.00250957x^{6.3}y^{5.75} +$$

$$9.31323e^{-10}x^{1.2}y^2 + 0.0564374x^{3.5}y^{3.75} - 0.00954771x^{3.8}y^{3.5}$$

$$-0.0000386088x^{7.8} y^{7.5}$$

*If n = 1, we get*

$$u_2(x,y) = 7.27596e - 12x^{5.5}y^{5.75} + 0.000353838x^{5.8}y^{5.5}$$

$$-0.00000111428x^{8.1}y^{7.25} - 0.0000562515x^{6.1}y^{5.25} - 4.65079e - 8x^{4.3}y^{3.75}$$

$$-0.00000498821x^{8.6}y^{7.5} + 1.0x^{2.0}y^2 - 0.00250957$$

$$x^{6.3}y^{5.75} - 0.0000841583x^{6.6}y^{5.5} - 2.42514e^{-15}x^{1.2}y^2$$

$$+5.82077e^{-11}x^{3.5}y^{3.75} - 4.94072e^{-8}x^{9.6}y^{9.25}$$

*If n = 2, we get*

$$u_3(x,y) = -4.89841e^{-9}x^{10.9}y^{9.25} - 7.20333e^{-12}x^{5.5}y^{5.75} + 2.27374e^{-13}$$

$$x^{5.8}y^{5.5} + 8.25388e^{-7}x^{8.1}y^{7.25} - 1.61021e^{-7}x^{8.9}y^{7.25} - 7.27596e^{-12}x^{4.3}$$

$$y^{3.75} - 0.00000498821x^{8.6}y^{7.5} + 1.0x^{2.0}y^2 - 0.00250957x^{6.3}y^{5.75}$$

$$-3.03032e^{-10}x^{6.6}y^{5.5} - 1.25876e^{-7}x^{8.4}y^7 - 1.7528e^{-21}x^{1.2}y^2 -$$

$$1.24008e^{-9}x^{10.4}y^9 - 3.40038e^{-11}x^{11.9}y^{11} + 1.77679e^{-14}x^{7.8}y^{7.5}$$

*Figure 2 compares the exact and the approximated solutions for different values of y.*

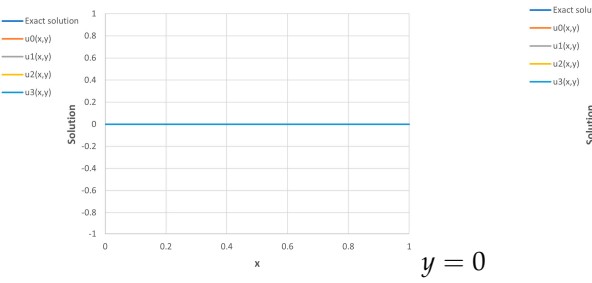
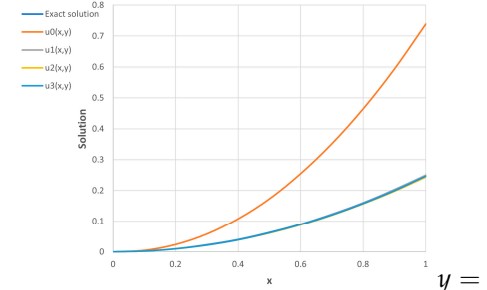

**Figure 2.** *Cont.*

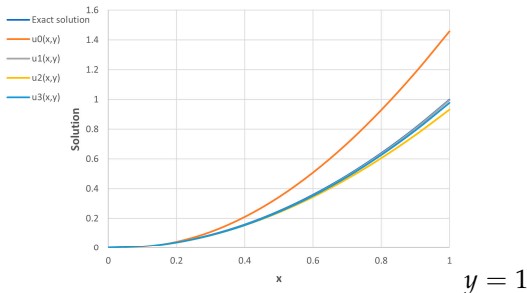

**Figure 2.** Exact and approximate solutions of Example 2.

### 6. Conclusions

The present study shows that VIM is a very accurate method that gives the exact solution in a few steps. In some cases, however, it requires more calculations, which will add some difficulties to the problem under consideration. This work may be improved in future by including integro-differential equations with kernels including fractional-order derivatives of the unknown function, in addition to considering fractional-order derivatives greater than 1.

**Author Contributions:** Conceptualization, M.D. and Z.A.M.; methodology, M.D.; software, Z.A.M.; validation, M.D. and Z.A.M.; formal analysis, M.D.; investigation, Z.A.M.; resources, Z.A.M.; data curation, Z.A.M.; writing—original draft preparation, M.D.; writing—review and editing, Z.A.M.; visualization, Z.A.M.; supervision, M.D.; project administration, M.D. All authors have read and agreed to the published version of the manuscript.

**Funding:** This research received no external funding.

**Institutional Review Board Statement:** Not applicable.

**Informed Consent Statement:** Not applicable.

**Data Availability Statement:** Not applicable.

**Conflicts of Interest:** The authors declare no conflict of interest.

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
