# Peer review of "Variational Iteration Method for Solving Fractional Integro-Differential Equations with Conformable Differointegration"

_axioms, doi:10.3390/axioms11110586_

Round 1

Reviewer 1 Report

In the article under review, the authors used the variational iteration method to solve the fractional integro-differential equations with conformable differointegration. They derived the iterative sequence of approximate solutions by using the variational iteration method and then proved that the convergence of this sequence to the exact solution under certain conditions over the kernel of the integro-differential equation. Two examples were provided to show the effectiveness of the variational iteration method for solving the linear and nonlinear integro-differential equations with conformable fractional order differointegration.

After reading and checking the full article, I didn't find any mathematical and logic mistakes. The proposed methods and techniques are interesting for readers who are working on the related fields. Therefore, I recommend the article to be accepted for publication in the Axioms after providing minor revision.

Comments and suggestions:

(1) The authors should read and check the full article very carefully to correct possible grammar and spelling mistakes;

(2) Rewrite the Abstract section to make it more details and precise;

(3) Further strengthen the motivation for writing this article in the Introduction section;

(4) Polish the English writing of the full article;

(5) Some published articles focus on the topic of the article are missing in the references. I suggest the authors to cite the following three items in the text and in the references such that the readers can better understand the latest progress in this research field:

https://doi.org/10.1016/j.camwa.2009.03.048

https://doi.org/10.1002/mma.7954

https://doi.org/10.1515/math-2021-0093

Author Response

Dear reviewer

1.we are check the full article very carefully and we correct every grammar and spelling.

2. we are rewrite the abstract section and we make it more details of objectives.

3.we are made the motivation for writing this article in the Introduction section and we added more references to support introduction that you suggested.

4.you will see polish the english writing of the full article.

finally , we are use the articles suggested that focus on the topic and we make  to cite it.

l

Reviewer 2 Report

Dear Editor,

I have read the paper with the title “Variational Iteration Method for Solving Fractional Integro-Differential Equations with Conformable Differointegration” in details. The manuscript is written well. There are many useful results in the paper. I suggest it for publication after the following changes.

-There are some grammatical errors and typos. The authors should correct all.

-The introduction section should be rearranged.

-The motivation of the paper should be presented in details.

-The references list should be updated. I suggest the following works.

Numerical Investigation of Nonlinear Shock Wave Equations with Fractional Order in Propagating Disturbance, Symmetry 14 (6), 1179

Novel Mathematical Modelling of Platelet-Poor Plasma Arising in a Blood Coagulation System with the Fractional Caputo–Fabrizio Derivative, Symmetry 14 (6), 1128

Approximate Solution of Nonlinear Time-Fractional Klein-Gordon Equations Using Yang Transform, Symmetry 14 (5), 907

Two Dimensional Laplace Transform Coupled with the Marichev-Saigo-Maeda Integral Operator and the Generalized Incomplete Hypergeometric Function, Symmetry 13 (12), 2420

Author Response

Dear reviewer

1.we are improved the grammatical to the full paper and we correct all errors.

2.we arranged the introduction section as shown in the attached file.

3. we are making the motivation of the paper and we presented in detail.

4.we use the articles proposed and added to references paper.  

Reviewer 3 Report

In this article, by utilizing variational Iteration Method, the authors discussed the sequences solutions of approximation and the exact solution for solving fractional integro-differential equations with conformable dfferointegration in real Banach spaces. The authors presented theorems and corresponding proofs. The authors also presented some examples. The results are interesting.

1.    Theory and application of nonlinear fractional differential equations, such as

https://doi.org/10.1016/j.amc.2015.10.020

I suggest that authors cite the paper and in the introduction to enrich the literature review of this paper.

2.    I suggest the authors that point out the prospect and development of the paper research in conclusions .

3.The paper has some typos. The authors should carefully check the full text and correct it.

The manuscript can be considered for publication provided that the authors revise the paper base on the above suggestions.

Author Response

Dear reviewer 

1.we are added the suggested that cite the paper to improve the introduction of paper and added to references.

2.edited and developed the conclusions of the paper and we suggested future work.

3.we made a correction all typos, to appear in best shape.  
